Impact assessment of high soil CO2 on plant growth and soil environment: a greenhouse study

He Wenmei
Yoo Gayoung gayoo@khu.ac.kr
Moonis Mohammad
Kim Youjin
Chen Xuanlin
Department of Applied Environmental Science, Kyung Hee University , Yongin-si , South Korea
Abiven Samuel
Electronic publication date: 2019 Jan 25
Publication date: 2019
Volume: 7
Electronic Location ID: e6311
Received 2018 Oct 11; Accepted 2018 Dec 19
Copyright: ©2019 He et al.
Copyright year: 2019
Copyright holder: He et al.
License: This is an open access article distributed under the terms of the Creative Commons Attribution License, which permits unrestricted use, distribution, reproduction and adaptation in any medium and for any purpose provided that it is properly attributed. For attribution, the original author(s), title, publication source (PeerJ) and either DOI or URL of the article must be cited.
License URL: https://creativecommons.org/licenses/by/4.0/

Keywords: CO2 stress, High soil CO2, Carbon capture and storage, Root water absorption activity, Plant response, Impact assessment, O2 depletion

Funding: Korea Ministry of Environment Korea-CO 2 Storage Environmental Management (K-COSEM) Research Program 2014001810002 This work is supported by the Korea Ministry of Environment (MOE) as the Korea-CO2 Storage Environmental Management (K-COSEM) Research Program (Project No. 2014001810002). The funders had no role in study design, data collection and analysis, decision to publish, or preparation of the manuscript.

==============================
To ensure the safety of carbon capture and storage (CCS) technology, insight into the potential impacts of CO2 leakage on the ecosystem is necessary. We conducted a greenhouse experiment to investigate the effects of high soil CO2 on plant growth and the soil environment. Treatments comprised 99.99% CO2 injection (CG), 99.99% N2injection (NG), and no injection (BG). NG treatment was employed to differentiate the effects of O2 depletion from those of CO2 enrichment. Soil CO2 and O2 concentrations were maintained at an average of 53% and 11%, respectively, under CG treatment. We verified that high soil CO2 had negative effects on root water absorption, chlorophyll, starch content and total biomass. Soil microbial acid phosphatase activity was affected by CG treatment. These negative effects were attributed to high soil CO2 instead of low O2 or low pH. Our results indicate that high soil CO2 affected the root system, which in turn triggered further changes in aboveground plant tissues and rhizospheric soil water conditions. A conceptual diagram of CO2 toxicity to plants and soil is suggested to act as a useful guideline for impact assessment of CCS technology.

Introduction

Carbon capture and storage (CCS) technology is considered an important strategy for mitigating climate change (IPCC, 2014). To ensure safe and successful CCS projects, the European Union has published guidance documents for its CCS Directive that include general CO2 leakage scenarios (European Communities, 2011; Pearce et al., 2014). The most relevant CO2leakage could occur in storage reservoirs via faults, fractures or boreholes (Pearce et al., 2014; Vrålstad et al., 2018). As CO2 would likely be transported to storage sites through long pipelines, leakage could also occur via corrosion and connection failures of shallow pipes (European Communities, 2011; Fu & Gundersen, 2012; Pearce et al., 2014). The leaked CO2 would diffuse from deep soil layers toward the surface. As a result, soil could experience high CO2 concentrations between 40% and 95% for a period of time (Beaubien et al., 2008; Al-Traboulsi et al., 2012; Lake et al., 2013; Paulley et al., 2013). Surrounding plants and soil microbes could be influenced by high concentration of soil CO2 and the concomitant low pH and reduced proportions of O2 andN2.

Plant responses to a high-CO2 soil environment have been widely investigated in natural analogues and at artificial CO2 release sites (Pfanz et al., 2004; Beaubien et al., 2008; Male et al., 2010; Krüger et al., 2011; Patil, 2012; West et al., 2015). Chlorosis and discoloration were observed in natural vegetation after 4 days of CO2 exposure at the Zero Emission Research and Technology Center in the US (Lakkaraju et al., 2010; Sharma et al., 2014). Plant physiological indicators such as chlorophyll content, photosynthesis rate, stomata conductance and transpiration rate were lower in soils with a high CO2 concentration (Spangler et al., 2009; Wu et al., 2014). Morphological indicators such as plant height, root length, leaf number, leaf area, seed number and pod number were also reduced after the soil was exposed to high levels of CO2 compared with non-gassed controls. Plant biomass also decreased on CO2 gassing (Al-Traboulsi et al., 2012; Al-Traboulsi et al., 2013; Wu et al., 2014).

Although previous studies consistently reported that plants exposed to high soil CO2 showed inhibited growth (Patil, Colls & Steven, 2010; Al-Traboulsi et al., 2013; Zhao et al., 2017), few clearly identified the main driving factor of these negative impacts (Pfanz et al., 2004; Pfanz & Saßmannshausen, 2008; He et al., 2016). Beaubien et al. (2008) and Stephens & Hering (2002) observed that water absorption of plants was lower, but they did not distinguish the effects of high soil CO2 (∼100% at 20 cm depth) from those of low O2 or other trace gases such as H2S and CH4. Al-Traboulsi et al. (2012) and Zhang et al. (2016) explained that reduced bean and maize metabolism in CO2 gassing plots might be due to reduced O2 level rather than pH changes induced by high soil CO2 concentration (50–70% at 15–30 cm depth). Lake et al. (2016a) sought to distinguish the effects of elevated CO2 (42.3%) from those of O2 depletion (11.1%) on the growth of wheat and beetroot by employing an N2 gassing treatment. They suggested that high soil CO2 might explain the biomass reduction of beetroots better than O2 deficiency. Similar results were found in earlier studies by Chang & Loomis (1945) and Kramer (1940). In their studies, transpiration and root water absorption of wheat, maize, rice, sunflowers and tomatoes were close to normal under O2-free nitrogen-bubbling treatment, while they were severely affected by 100% CO2 bubbling. Chang & Loomis (1945) also compared the effects of low pH solution and high CO2 exposure on root function and observed that toxic effects were more evident under high CO2 conditions than under acidic solutions. They suggested that CO2 toxicity is a more important factor in plant growth than O2 deficiency. Previous studies have implied that high concentration of soil CO2 itself might be toxic to plant growth in a variety of plant species. However, compared with the intensive studies of CO2 effects on plants in hydroponic system, the assumption of CO2 toxicity on plant growth under high soil CO2 condition has not been sufficiently proved.

To identify the main factor in soil CO2 enrichment that influences plant growth and soil, we conducted a greenhouse experiment using a specially designed pot in which soil CO2 was enriched without a buildup of ambient atmospheric CO2. We separated the effects of high soil CO2 and those of low soil O2 by employing an N2 gassing treatment. The pot was large enough to grow woody plants, and the injection time covered the main growing period. The objective of this study was to investigate the effects of high soil CO2 on plant physiological reactions and their interactions with the soil environment. A conceptual diagram of the effects of CO2 toxicity on plant metabolism is proposed for further insight into the impacts of high soil CO2.

Materials & Methods

Soil and plant preparation

Soils used for incubation included commercial potting soil (Korea Association of Seedbed Media, Seoul, South Korea) and mineral soil procured from a construction company (Seung Hwa Construction, South Korea). Basic soil physicochemical properties and components are summarized in Table 1. Soil pH was measured using a 1:5 dilution method (Thomas, 1996). Total carbon (TC) and total nitrogen (TN) contents were analyzed using combustion analysis with a Carlo Erba NS 1500 C/N analyzer (Carlo Erba, Milan, Italy). A two-year-old crimson glory vine (Vitis coignetiae) was selected for this experiment and bought locally (West Suwon Agricultural Products Inc., South Korea). This plant grows well in neutral to alkaline soils and is native to East Asian countries such as Japan and South Korea.

Table 1 Physicochemical properties of the soil.

Soil	pH	TN	TC	Composition (%)	
		(g kg−1 soil)	Zeolite	Pearlite	Vermiculite	Coco peat	Peat moss	Other	
Potting	6.87	6.30	390.32	4.0	7.0	6.0	68.0	14.7	0.3	
				Clay	Silt	Sand				
Mineral	6.80	0.19	3.02	4.0	26.0	70.0				

Experimental design

Gas treatments were applied in a greenhouse on the campus of Kyung Hee University (Yongin-si, South Korea) for 32 days from August 20 to September 23, 2015. Mean temperature was 28 ± 4 °C during the day and 18 ± 5 °C at night. An acrylic container including three pots (30 × 20 × 20 cm; height × length × width) and an empty chamber (20 × 20 × 60 cm; height × length × width) were specially designed (Fig. 1A). At the bottom of each pot, nine drainage holes were covered with mesh for optimal gas diffusion. An empty chamber was connected to a gas tank through a valve. Each pot was filled with 7 kg of mineral soil up to 15 cm from the bottom, and then another 15 cm of potting soil was added. Grape saplings were transplanted on July 15 and adjusted to a new environment for 36 days until August 20. Soil water content in all treatments was adjusted to 20–30% (v/v) and maintained by manual watering every 4 days. In the CG treatment, 99.99% CO2 gas was supplied at a rate of 400 mL/min−1. The NG treatment was set up using 99.99% N2 gas at the same injection rate to investigate the effects of O2 depletion without CO2 enrichment. The BG treatment was prepared as a control with no injection. Three silicon tubes (2 mm in diameter) with one end connected to a copper fitting covered with mesh (100 µm) were buried in the soil at depths of 5, 10 and 20 cm to measure the CO2 and O2 concentrations. Although the pot was not large enough to permit lateral heterogeneity of the gases, the tubes at each depth in the replicated pots were located randomly to ensure measurements were representative. Nine replicates were set up for each treatment. The greenhouse remained open and two fans were used to prevent ambient CO2 accumulation (Fig. 1B).

Figure 1 Injection box (A) and treatment layout in the greenhouse (B).

Photo credit: Wenmei He.

Soil measurements

Daily measurements of soil CO2 and O2 (% of total 100% by volume) at three depths (5, 10, and 20 cm) were performed using a portable gas detector (COMBI-R, Status Scientific Controls Ltd., Mansfield, Nottinghamshire, UK). The detection range of CO2 was from 0 to 100% with an accuracy of ±1% and that of O2 was from 0 to 25% with an accuracy of ±0.1%. Soil pH and temperature were measured directly using a pH meter (Field Scout pH 600; Spectrum Technologies, Inc. Plainfield, IL, USA). Soil volumetric water content was measured using a moisture sensor (GS1, Decagon Devices, Inc., Washington D.C., USA) at 5 cm depth. Plant root water absorption activity (RWAA) was measured indirectly by monitoring changes in soil water content in each pot. Soil with active root metabolism tends to lose more water when soil water change occurs only through plant transpiration (Stephens & Hering, 2002). We assumed an equal amount of water drainage from pots and surface soil evaporation among the different treatment pots (Allen et al., 1998). Hence, we defined RWAA as the daily amount of water loss during the no watering period. A larger RWAA indicated greater root activity.

Samples (200g) of potting and mineral soils were removed from each pot at the end of the injection period and stored at 4 ° C before analysis. Enzyme analysis was conducted within one week of sampling. Five enzyme activities were determined by fluorometric assays using methylumbelliferone linked substrates, including acid phosphatase (AP), cellobiohydrolase (BC), 1,4-β-glucosidase (BGC), xylosidase (BX) and 1,4-β-N-acetyl glucosaminidase (NAG), which are enzymes that mediate key functions during microbial degradation of soil litter (Chung et al., 2011). AP cleaves phosphoester bonds; BC and BGC decompose cellulose; BX breaks down hemicellulose and NAG degrades chitin (Saiya-Cork, Sinsabaugh & Zak, 2002). The TC and TN contents of air-dried soils were analyzed with a Carlo Erba NS 1500 C/N analyzer (Carlo Erba, Milan, Italy). The hot water extractable carbon (HWC) of the potting and mineral soils, which is a proxy for the amount of labile carbon, was determined following the method described in Nelissen et al. (2012).

Plant measurements

Photographs were produced every week to record visible changes in plants, and chlorophyll a content was measured to monitor physiological changes. To measure chlorophyll a, leaves were sampled twice a week from August 20 to September 3 and once a week from September 4 to September 23. The leaves were homogenized by grinding with a mortar and pestle and chlorophyll a was extracted using 90% acetone overnight at 4 °C in the dark. A UV/Vis spectrophotometer (Optizen POP, Mecasys Co., Ltd, Daejeon, South Korea) was used to measure chlorophyll a concentration (Arnon, 1949).

When the CO2 injection period was completed, the plant was carefully dug out and separated into roots and shoots (leaves and stems). Root samples were washed with water and dried at 60 °C for 3 days to obtain dry weights. The fresh weight of the shoots was recorded, and a subsample of fresh shoots was dried to determine water content. To monitor movement and distribution of nutrients, TC, TN and total phosphorus (TP) contents of the plants were measured. The TC and TN contents of the roots and leaves were subjected to combustion analysis with a Carlo Erba NS 1500 C/N analyzer. TP content of leaves was measured using a modified molybdenum blue technique (Murphy & Riley, 1962). The starch content of plant roots was measured using a starch assay kit (K-TSTA, Megazyme International Ireland Co. Ltd., Wicklow, Ireland) as a stress indicator that decreases when plants are under stress (Lake et al., 2012).

Statistical analysis

Analysis of variance was performed using the MIXED procedure in SAS 9.1 (SAS Institute Inc., Cary, NC, USA) and included concentrations of soil CO2 and O2, pH, chlorophyll a content, and RWAA, while an additional GLM procedure was used to analyze plant parameters (biomass, starch, TC, TN, TP and water content) and soil parameters (TC, TN, HWC contents and enzyme activities). Least square means were used to test for significant differences among treatments at a 5% probability level.

Results

Soil gases and pH

Soil CO2 concentrations were significantly higher in the CG treatment than in the NG and BG treatments (Fig. 2, Tables 2 and 3). Due to gas exchange with the atmosphere, we observed a CO2 gradient at depths in the CG treatment (Fig. 2A and Table 2). The gas concentrations from locations of nine replicates had small deviations at each depth (Fig. 2), indicating that lateral heterogeneity of gases could be ignored in our small pots. Constant CO2 concentrations were maintained throughout the injection period indicating that our incubation system was sufficiently stable to investigate the effects of high soil CO2 on plant growth. In the BG and NG treatments, soil CO2 concentrations were lower than 1%, which is below the detection threshold (Table 2). These concentrations were comparable with those of field studies that reported <1% CO2 in the non-injection plots (0.7–0.9%) measured by high-accuracy gas analyzers (GA5000 and Vaisala 221 GMT probe) (Sharma et al., 2014; Lake et al., 2016a). Soil O2 concentrations in the BG treatment were similar at all depths and the same as that at the ambient condition (Table 2). In the NG and CG treatments, soil N2 and CO2 injections significantly reduced O2 concentrations, which were maintained at 7.4–14.3% throughout the injection period (Figs. 2B–2C and Table 2). Soil CO2 and O2 concentrations were negatively correlated in the CG treatment (Fig. 2D). Soil pH was slightly increased at the end of the injection period compared with the initial pH (Tables 1 and 2), and the CG treatment significantly reduced soil pH compared with pH in the BG and NG treatments (Table 2).

Figure 2 Mean soil CO2 (A) and O2 (B) concentrations in CG treatment. Mean soil O2 concentration in NG treatment (C). The relationship between soil CO2 and O2 concentrations in CG at each depth (n = 256) (D)

Bars with the same letters are not significantly different at a 5% level (n = 9).

Table 2 Mean soil CO2 and O2 concentrations in different depths and soil pH at each treatment.

Treatments		CO2 concentration (%)	O2 concentration (%)	pH	
	Depth (cm)	5	10	20	5	10	20		
CG		41.3a	53.8a	65.3a	14.3b	11.8b	8.4b	7.4a	
NG		<1.0b	<1.0b	<1.0b	14.1b	10.3b	7.5b	7.5a	
BG		<1.0b	<1.0b	<1.0b	21.0a	20.9a	20.9a	7.0b	
Notes.

a,b Different letters indicate significant differences among treatments at a 5% level (n = 261).

Table 3 Analysis of variance examining the effects of CO2 injection on soil parameters.

Source	CO2	O2	pH	Chlorophyll a	RWAA	
Date	<0.0001	<0.0001	<0.0001	<0.0001	<0.0001	
Depth	<0.0001	<0.0001	–	–	–	
Date × Depth	0.0004	<0.0001	–	–	–	
Treatment	–	<0.0001	<0.0001	<0.0001	<0.0001	
Date × Treatment	–	<0.0001	<0.0001	0.9941	0.0512	
Depth × Treatment	–	<0.0001	–	–	–	
Date × Depth × Treatment	–	<0.0001	–	–	–	
Source	Soil	
	Potting	Mineral	Microbial enzyme activities	
	TC	TN	HWC	TC	TN	HWC	AP	BC	BGC	BX	NAG	
Treatment	0.6421	0.2532	0.983	0.8781	0.0298	0.0028	0.0042	–a	0.2503	0.0174	<0.0001	
Notes.

a No data.

Plant parameters during injection

Temporal changes in leaf color and vitality varied by treatment (Fig. 3). In the BG treatment, plant leaves remained healthy and green until the end of the experiment (Fig. 3A). In the NG treatment, plant leaves appeared healthy until September 8 (Fig. 3B); after that, a few leaves at the bottom of the plant turned yellow. In the CG treatment, plant leaves appeared healthy until September 8 and then turned yellow and red. At the end of experiment, the leaves in the CG treatments appeared very dry and showed low vitality (Fig. 3C).

Figure 3 Morphological changes of plants during the experimental period.

(A) The photographs of grape in BG (no injection) from left to right were taken on August 21, September 8 and September 22; (B) The photographs of grape in NG (N2 injection) from left to right were taken on August 21, September 8 and September 22; (C) The photographs of grape in CG (CO2 injection) from left to right were taken on August 21, September 8 and September 22. Photo credit: Wenmei He.

In all treatments, chlorophyll a content increased quickly until August 25 and decreased slowly thereafter, following the seasonal night temperature drop in all the treatments (Table 3 and Fig. 4A). Treatment effects varied by date and became apparent after the one-week injection. On August 28, September 8, and September 16, the chlorophyll a content of leaves in the CG treatment was significantly lower than that in the BG treatment. However, chlorophyll a in the NG was not significantly lower than in the BG on any date of measurement.

Figure 4 Leaf chlorophyll a content (A) and root water absorption activity (RWAA) (B).

Each date was compared and results with the same letter are not significantly different at a 5% level (n = 9).

An overall decreasing pattern in RWAA was observed in all the treatments (Table 3 and Fig. 4B). This could be due to seasonal temperature drop, which leads to slower plant metabolism. Treatment effects became apparent from August 24, when RWAA in the CG treatment was significantly lower than in the BG and NG treatments. This effect continued until the end of the experiment. RWAA was not significantly different between the NG and BG treatments. The effects of high soil CO2 on RWAA were observed 4 days earlier than those on chlorophyll a content.

Plant parameters after harvest

After injection, shoot and root biomass were reduced by 19.2% and 59.2%, respectively, in the CG treatment compared with the BG treatment (Table 4). Shoot and root biomass, however, were not changed in the NG compared to the BG. Water content of the shoots (leaves and stems) and starch content of the roots were also significantly lower in the CG treatment, but they were not significantly different between BG and NG treatments. TC content in leaves and roots was similar between BG and CG treatments. TN content was higher in leaves and roots in the CG treatment compared with the BG and NG treatments. The TP content of leaves was lower in the NG and CG treatments than in the BG treatment (Table 4).

Table 4 Total shoot and root biomass, shoot water, root starch, TC and TN contents in leaves and roots.

	Total biomass	Shoot biomass	Root biomass	Shoot water content	Root starch	TC	TN	TP	
						Leaf	Root	Leaf	Root	Leaf	
	(g plant−1)	(%)	(g kg−1)	
BG	58.22a	34.36a	23.87a	57.37a	16.75a	435.78a	473.89a	17.94a	4.80b	5.26a	
NG	56.50a	35.88a	20.62a	55.57a,b	17.55a	435.82a	464.78b	15.83b	4.43b	3.80b	
CG	37.48b	27.75b	9.73b	48.68b	2.25b	436.42a	482.22a	19.62a	7.89a	4.18b	
P	<0.0001	0.0176	<0.0001	0.0754	<0.0001	0.9739	0.0019	0.0970	<0.0001	0.0238	
Notes.

a,b Different letters indicate significant differences among treatments at a 5% level (n = 9).

Soil parameters after experiments had finished

Soil TC and TN contents did not vary among treatments, while HWC contents in mineral soil were lower in CG than in the BG (Table 5). On the other hand, enzyme activities in potting soil were different among treatments. The activity of AP was lower in the CG and NG than in the BG, while activities of NAG and BX were higher in the NG than in the BG and CG (Table 5). In mineral soils, enzyme activities were too low to be detected.

Table 5 TC, TN and HWC contents and microbial enzyme activities in soil.

Parameters	Soils	BG	NG	CG	
			Potting	Mineral	Potting	Mineral	Potting	Mineral	
Chemical parameters	TC	(g kg−1)	250.96a	3.38a	224.74a	3.35a	257.76a	3.22a	
TN	4.01a	0.68a	4.96a	0.48a,b	5.17a	0.36b	
HWC	4.62a	0.59a	4.73a	0.57a,b	4.67a	0.55b	
Microbial enzyme activities	APc	(nmol h−1 g−1)	200.56a	-d	114.40b	–	86.98b	–	
BC	–	–	7.93	–	–	–	
BGC	83.75a	–	75.16a	–	50.12a	–	
BX	5.32b	–	10.20a	–	5.38b	–	
NAG	27.30b	–	75.22a	–	14.63b	–	
Notes.

a, b Different letters indicate significant differences among treatments at a 5% level (n = 9).

c AP, BC, BGC, BX and NAG stand for: acid phosphatase, cellobiohydrolase, 1,4-β-glucosidase, xylosidase, and 1,4-β-N-acetyl glucosaminidase, respectively.

d No data.

Discussion

The results clearly indicated that injected gases migrated upwards through all the gassed pots. A clear depth gradient of soil CO2 was observed in these pots, while lateral heterogeneity in CO2 was negligible (Fig. 2). High soil CO2 concentration lowered soil O2 and pH, consistent with the previous results. The negative relationship between soil CO2 and O2 concentrations in the CG could be due to O2 displacement by injected CO2 (Patil, 2012; Lake et al., 2016a). The steeper slope in the linear negative relationship between CO2 and O2 at 5 cm depth than those at 10 cm and 20 cm (Fig. 2D) implied that, in the shallow surface layer, diffused CO2 gas was mixed with atmospheric air (Fig. 2A). These vertical gradients in soil CO2 concentration were also observed in other greenhouse and field studies (Krüger et al., 2011; Wu et al., 2014; West et al., 2015; Kim et al., 2018). In addition, our range of CO2 concentrations of 41.3–65.3% at 5–20 cm depths (Table 2) was comparable to the levels in previous studies (Beaubien et al., 2008; Al-Traboulsi et al., 2012; Lake et al., 2013; Lake et al., 2016a), and this the possible level of CO2 concentration at which CO2 from the CCS site would leak and be transported via faults or failure.

The reductions in chlorophyll a content, RWAA and total biomass seen with CG treatment and not in the NG treatment indicate that plants were influenced by high soil CO2 but not by low O2 concentration. Although CO2-induced reduction in soil pH in the CG treatment could affect plants, we put little emphasis on the effect of lowered pH. In this experiment, soil pH in the CG treatment (7.0) was lower than in BG and NG (pH = 7.4 on average) (Table 2), but the difference was only 0.4, which is within the optimal range for grape growth. The possible reason for such a slight change in soil pH despite CO2 gassing, which could lead to formation of weak acid (H2CO3) and release of H+ (Wei, Maroto-Valer & Steven, 2011; Moonis et al., 2017; Zhao et al., 2017), could be the high buffering capacity of the potting soil and plants’ indirect effects. Our results were consistent with those of Lake et al. (2013), who reported that beetroots were not significantly affected by N2 gas treatment at O2 concentration of 8.7%. Even in an O2-free hydroponic solution, Boru et al. (2003) and Chang & Loomis (1945) did not observe inhibited root growth or function, implying that plants could be resistant to low O2 condition (Geigenberger, 2003). Nikolopoulou et al. (2012) reported clear results that grass (Cynodon dactylon) root length and biomass were greatly reduced (>50%) under treatment with 40% CO2 combined with 20% O2, and a similar reduction was observed under treatment with 40% CO2 combined with 10% or 2.5% O2. We therefore suggest that high soil CO2 concentration has negative effects on overall plant metabolism.

We observed the effects of high soil CO2 concentration on plants with regard to root function. The reduction in RWAA in the CG treatment was apparent on August 24, 4 days after the start of injection (Fig. 4B). On August 28th, lower chlorophyll a content was first observed in the CG treatment (Fig. 4A). Leaf color change became apparent as late as September 8 (Fig. 3C). The series of changes in the CG treatment imply that high soil CO2 concentration affected the root system first, which triggered other aboveground changes in the plant. Color changes in the leaves are related to changes in nutrient transport (Chang & Loomis, 1945; Cook, Ward & Wicks, 1983; Boru et al., 2003; Geigenberger, 2003). In our study, the TN content of the roots was higher in the CG than in the BG treatment, while there were similar TN and lower TP contents in the CG plant leaves compared with the BG leaves (Table 4). These results indicated that nitrogen transport from the roots to the leaves was inhibited by high soil CO2 concentration.

CO2 injection also induced changes in soil HWC content and microbial enzyme activities. Reduced HWC in mineral soil of the CG treatment could be related to the high soil CO2 concentration (Table 5). Moonis et al. (2017) reported that soil CO2 injection increased exchangeable Al3+ of mineral soil, which would be strong bound with dissolved C molecules. The resulting Al3+-C complexes could be absorbed on the soil surface, producing reduced water-extractable C. Reduced soil microbial enzyme activity of AP in the CG and NG (Table 5), which is involved in cleaving phosphoester bonds, could be related to the decreased TP contents of the leaves in those treatments (Table 4). The increased enzyme activities of NAG and BX in the NG treatment (Table 5), which are involved in decomposition of hemicellulose and chitin, respectively, might be related to lower soil O2 concentration. In a low O2 environment, the activity of N2-fixation organisms could be enhanced, resulting in increased soil-available N. This change could be beneficial to microbial decomposition processes that are usually limited by the low N content of residues in soil (Rice & Paul, 1972; Halsall & Gibson, 1985; Chen et al., 2011).

Figure 5 Conceptual model of the response to high soil CO2 concentration in plant: involvement of hydraulic reaction and compensational mechanism.

Photo credit: Wenmei He.

A mechanism for CO2 toxicity to plant root tissue was suggested by earlier studies (Chang & Loomis, 1945; Bown, 1985). High soil CO2 concentrations could dramatically change the pH level of root tissues, which could undermine membrane function and osmoregulation of root cells. Damaged root cells would not absorb water sufficiently, and plant stomata would close to prevent water loss (Davies, Kudoyarova & Hartung, 2005). Similarly, Lake et al. (2016b) suggested a signaling mechanism from the root to the leaf: when plant roots suffer CO2 stress, they produce hormones such as abscisic acid and send this signal to the leaves to close stomata. As a compensatory mechanism, the plant hydrolyzes starch in the roots and stems into soluble sugars and changes the osmotic potential to increase water absorption (Yang et al., 2001; Mohammadkhani & Heidari, 2008; Naser et al., 2010). In our study, the reduced starch content of roots in the CG treatment (Table 4) could be associated with this compensation response to high soil CO2 stress (Nikolopoulou et al., 2012). Based on our observations and extensive literature review, we propose a conceptual diagram of plant response to high soil CO2 concentration (Fig. 5). The effects of high soil CO2 on plants start from the root cells where intercellular pH is reduced, leading to activation of signaling to the leaf to close the stomata. Due to stomata closure, water and nutrients are not transported from the soil to the leaves, resulting in low photosynthesis. Although plants try to compensate for lower photosynthates by solubilizing starch, they eventually consume stored resources, lose vitally and die.

Conclusion

Plants can be a useful tool for assessing the potential risk of CO2 leakage from CCS sites. We verified that high soil CO2 (41.3–65.0%) had negative effects on overall plant metabolism. These negative effects were attributed to high soil CO2 concentration and not to low soil O2 concentration or to reduced soil pH. Negative effects on root water absorption were observed after CO2 injection, and the effects extended to aboveground plant tissues, leading to decreased chlorophyll content and chlorosis.

Results from our study suggest two sensitive indicators for CO2 leakage in ecosystem. The first indicator is chlorophyll content which showed a reduction 8 days after CO2 injection. Another sensitive indicator was RWAA, which showed reduction 4 days earlier than chlorophyll content. However, as RWAA is difficult to measure in the field, soil water content could be used as an indirect parameter for detecting CO2 leak. The inherent heterogeneity of soil water content in space and time could be overcome by establishing long-term baseline data in a given site. Suggested indicators could be utilized to develop a guideline for environmental management of CCS and the surrounding environment. Further research should focus on the responses of different plant species to high soil CO2 and the interactions between plants and soil.

Supplemental Information

Supplemental Information 1 Soil CO2 concentration

Click here for additional data file.

Supplemental Information 2 Soil O2 concentration

Click here for additional data file.

Supplemental Information 3 Chlorophyll a contents

Click here for additional data file.

Supplemental Information 4 Soil moisture

Click here for additional data file.

Supplemental Information 5 Soil pH

Click here for additional data file.

We would like to acknowledge Konkuk University for its supports of analysis of enzyme activities in this study. We would also like to thank all participants for their full cooperation during the experimental period.

Additional Information and Declarations

Competing Interests

Author Contributions

Data Availability

The authors declare there are no competing interests.

Wenmei He conceived and designed the experiments, performed the experiments, analyzed the data, prepared figures and/or tables, authored or reviewed drafts of the paper, approved the final draft.

Gayoung Yoo conceived and designed the experiments, performed the experiments, contributed reagents/materials/analysis tools, authored or reviewed drafts of the paper, approved the final draft.

Mohammad Moonis conceived and designed the experiments, performed the experiments, analyzed the data, prepared figures and/or tables, authored or reviewed drafts of the paper.

Youjin Kim and Xuanlin Chen performed the experiments, analyzed the data.

The following information was supplied regarding data availability:

The raw measurements are available in the Supplemental Files.

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
