# Peer review of "Impact assessment of high soil CO2 on plant growth and soil environment: a greenhouse study"

_PeerJ, doi:10.7717/peerj.6311_

## Round 0.1 · original submission · Major Revisions

Your article was well received by both reviewers, but reviewer 1 proposes to go deeper into the data and the discussion. I would be grateful if you can address these comments.

Reviewer 1 ·

Basic reporting

1. Basic Reporting
English has to be revised by a native speaker, there are some edits and some sentences are difficult to read.
Some references are missing, especially in the introduction section; some suggestions are given in the ‘general comments’ section. Nevertheless, the ‘state of the art’ level of this article is acceptable.
The article is well structured ; some sub-sections are very short but this helps during reading.
The figures are relevant but they may be modified to increase readability :
- Figure 2 : the symbols are too large, sometimes it is not possible to see the error bars ; the scaling on the Y-axis has to be detailed to allow a better reading.
- Figure 4 : the labels a and b are not located appropriately : it is hard to understand to which curve the labels belong.
Raw data are available: it is suggested that the authors have a closer look to these data. For example, they do not consider the bivariate plot O2 vs CO2 (commonly used in soil gas monitoring dealing with CCS applications). By doing so, they may come to more in-depth description of the concentration changes during their experiment, and more specifically if some lateral heterogeneities exist. When using the data for CG container at 5 cm depth, one can find that the slopes and the correlation coefficents may differ from one sampling location to the other. This should be discussed.

Experimental design

This article meets the aims and scope of the journal. The research question is well defined, it has some interest to the scientific community dealing with CCS. Previous studies have also dealt with this subject but getting additionnal information is always useful. Nevertheless, some of the statements made in the introduction must be detailed or revised if needed. This is discussed in the ‘general comments’ section.
The ‘material and methods’ section is quite long (80 lines) compared to the discussion section (55 lines). However, some information is missing: if the equipment used during the monitoring is mentioned, there is no consideration about the accurracy nor the repeatability of the measurements. This is surprising. CO2 concentrations are only given at the percent level. This may be acceptable for high concentration range (more than 40-50% vol.), but this is not convenient for lower concentrations (O2 concentrations are given at 0.1%). The consequence is that there are no CO2 data available for NG and BG but only for CG. This has to be explained, because in studies performed in real conditions, a 1% rise of the CO2 concentration over a short time period may have some significance (e.g. link with diurnal cycle or link with CO2 intrusion). If the authors want to extrapolate their findings in the field, this has to be considered.

Validity of the findings

This article introduces new experiments to investigate the effect of high CO2 concentrations on plants. The monitoring is performed using different approaches (control plot, N2-enriched plot, CO2-enriched plot) with many monitoring points where the gas concentrations are measured regularly during one month. This is not the first time such an experiment is performed but the used geometry is appropriated and interesting results are obtained.
The authors have chosen to work with very high CO2 concentrations ; to my opinion this is one of the main debatable point in their approach. There are previous works (e.g. West et al., 2015, IJGCC 42, 357-371) that have established that a concentration of 10%-15% CO2 soil gas at 20cm depth is a threshold level for observing changes in plant coverage. Therefore one may wonder why the authors used a 40 to 60% CO2 concentration in the soil, depending on the depth the measurement is made. At such high concentration, it is known that changes will occur. It would have been interesting to work at CO2 concentrations closer to the threshold level in order to monitor the impact on plants, even if changes may have been weaker.
I would welcome a more detailed discussion on this choice of working with high CO2 concentrations.

Additional comments

Introduction section : there are several statements that must be revised.

Line 30 : leakage from a pipeline : it’s really surprising to make a first statement on pipeline leakage, because it is not the main issue when dealing of leakage possibilities and CCS. Surface infrastructure is, by nature, more easy to monitor and to instrument, and there are years of knowledge from the oil and gas industry. One of the main problem is rather the leakage trough borehole completions, old boreholes that were not well abandoned, and after leakage through faults and fractures. Moreover, I’m not sure the reference of West et al., 2011, makes a link between pipes and CO2 leakage, if you refer to this reference and the cited references.

Another statement is very surprising, in Line 32 : the authors link leakage through faults with the closure of the injection well. This is certainly a shortcut because leakage through faults may occur during storage activity or after (once the wells have been plugged), or never.

Line 34 : the references of Lewicki and Zhou are not cited appropriately : they do not correspond to leakage from faults after site closure, they correspond to near surface engineered leakages. That is fundamentally different.

Line 35 : please remember that when CO2 is injected in the soil, it does not only lower the O2 concentrations, it also lowers the N2 concentrations. This has probably no effect on plants, but this has to be mentioned to highlight that all the endmembers are taken into account.

Lines 37-38 : some references are missing : West et al., 2015, earlier mentioned. The authors may also refer to Krüger et al., 2011, IJGCC 5, 1093-1098.

Line 50 : the reference from Beaubien et al., 2008, is appropriated, but the study site (Latera) is also characterised by quite high levels of H2S ; this gas phase has also adverse effects on plants.

Line 65 : if the authors refer to some references, they may also add that the plant community is different in areas where CO2 concentrations is high in the soil.

Results section :
The CO2 gradient in the soi lis very high : the authors should add reference to other studies, to establish (or not) if such a gradient can occur naturally.
I wonder if a 0.4 pH unit difference is really meaningful – considering that pH values are often given with at ±0.1 unit (even if pH-meters can give 3 digits, this has not real significance in field applications).

Discussion :
I’ve no fundamental remark to make, because the results are in agreement with those of previous studies. Maybe the authors can comment on the discussion lines 226-230 : they discuss the presence/absence of O2 in the soil, they may also mention the role of O2 exisiting in the atmosphere.

Reviewer 2 ·

Basic reporting

1. Basic reporting: This is a timely report into the effects of potential CO2 leakage in the soil environment from CCS storage sites. Intro and background thorough and relevant, well structured. Figs relevant and of sufficient quality. Raw data was supplied and checked. Language is generally good, with a couple of specific points (see below)

Experimental design

2. Experimental design is acceptable, the research question provides novel findings of biochemical and metabolic changes in response to extreme CO2 in the soil environment. Methods are described adequately for replication.

Validity of the findings

3. Validity of findings is acceptable with robust data. Speculation of the potential metabolic biochemistry is useful and in context and conclusions are correct for data presented.

Additional comments

Specific comments:
Line 22 – suggests that the root system affected the surrounding soil environment, but this is not dealt with in the paper. If this refers to an increase in soil pH, this is not explained in the text. Otherwise it needs re-phrasing.
Line 28 – English would be better as ‘ To ensure safe and successful CCs projects’
Line 33 – ‘ would experience a high soil CO2’
Line 73 – a conceptual diagram is proposed, not ‘will be developed’
Lines 79 and 83 – South Korea
Line 172 – soil pH was statistically significantly reduced in CG treatment
Line 197 – should read after harvest, not after injection (ambiguous)
Line 207 – after experiments had finished, not after injection
Line 208 – HWC is lower in CG and NG than BG (Table 5) – clarify/explain
Line 217 – ‘plants were influenced by high soil CO2, but not by low O2’ - NG treatment had a very high NAG content compared to BG and CG – please explain this result
Line 256 – Lake et al. 2016b
Line 257 – ‘when plant roots suffer from CO2 stress’
Line 275 – ‘reduced soil pH’
Reference list requires subscript in CO2

Table 3 second section – statistical analysis refers to data presented in Table 4, suggest moving to bottom of Table 4.
Table 5, BCG is omitted from the abbreviations footnote
Figure 3 legend should explain what a), b) and c) are.

---

## Round 0.2 · accepted · Accept

Thank you for your revised version of the manuscript. You addressed all the comments we had.

# Reviewer 1 ·

Basic reporting

This is a revised version of a manuscript earlier submitted this year.
Comments and suggestions have been taken into account by the authors. The manuscript has been improved, thus it can be considered for publication.

Experimental design

OK

Validity of the findings

OK

Additional comments

OK